# Impact of One-Year Dietary Education on Change in Selected Anthropometric and Biochemical Parameters in Children with Excess Body Weight

**DOI:** 10.3390/ijerph191811686

**Published:** 2022-09-16

**Authors:** Kamilla Strączek, Anita Horodnicka-Józwa, Justyna Szmit-Domagalska, Tomasz Jackowski, Krzysztof Safranow, Elżbieta Petriczko, Mieczysław Walczak

**Affiliations:** 1Department of Pediatrics, Endocrinology, Diabetology, Metabolic Diseases and Cardiology of the Developmental Age, Pomeranian Medical University, 71-252 Szczecin, Poland; 2Department of Biochemistry and Medical Chemistry, Pomeranian Medical University, 70-111 Szczecin, Poland

**Keywords:** childhood obesity, lipid profile, nutrition, behavioural intervention, glucose metabolism

## Abstract

Obesity is regarded as a civilization disease that increases mortality and the risk of cardiovascular complications. In Poland, the prevalence of excess body weight in the paediatric population has been steadily increasing. The consequences of excess body weight in the developmental age population affect children’s health and destabilize their development. Appropriate dietary interventions are the main non-invasive methods of preventing and treating obesity. They should be aimed at the whole family, optimally with the use of simple tools such as the Healthy Eating Pyramid. Due to the increasing prevalence of excess body weight in the developmental age population and the problems with the treatment of this condition, studies were undertaken in order to determine the impact of a dietary intervention on anthropometric and biochemical parameters in children with excess body weight. A total of 68 (72.3%) children completed the study. Based on BMI SDS, 59 (86.8%) were diagnosed with obesity and 9 (13.2%) with overweight. After the completion of the one-year dietary educational program, a significant improvement in weight loss, waist and hip circumference, as well as the value of the WHtR index was demonstrated. There was also a significant increase in the percentage of muscle tissue and a decrease in the content of adipose tissue in the bodies of examined children. A significant improvement in the parameters of carbohydrate metabolism, and almost all parameters of lipid metabolism, except for total cholesterol. A significant (by 28.0%) reduction in the incidence of fatty liver was also noted. No influence of dietary education on arterial blood pressure was observed.

## 1. Introduction

The World Health Organisation has recognised obesity as a global range ailment since the 1990s [1]. According to data from a report by the WHO, between the years 1975 and 2016, the amount of people with obesity has tripled and is still rising at an alarming rate [2]. Globally, the problem of obesity affects also children and adolescents, which makes it one of the most important public health issues of the 21st century [3]. Studies show that the incidence of overweight and obesity in children and adolescents aged between 5 and 19 has drastically increased from 4% in 1975 to more than 18% in 2016 [4]. Similar observations have been made in Poland. Kułaga et al. [5,6] diagnosed overweight in 18.7% of boys and 14.1% of girls of school age (7–18), and 12.2% of boys and 15% of girls of preschool age (2–6).

In the majority of cases, obesity is a consequence of the ongoing upkeep of a positive energy balance as a result of incorrect lifestyle choices [7,8]. Easy access to calorie-dense foods, larger portion sizes, reduced physical activity and increased screen time effectively disrupt the energy homeostasis of the body [9]. Stańczyk et al. [10] show that children between 12 and 18 years of age with arterial hypertension and prehypertension consumed 20.0% more energy than their energy demand. Overweight and obesity carry many health threats and are causes of numerous disorders: from motor system disorders, impaired glucose tolerance, liver and cardiovascular system diseases, down to emotional and psychosocial problems [11,12].

Many specialists think that the treatment of obesity should consist of proper nutrition, physical activity, psychological therapy, pharmacotherapy and in some cases surgical intervention. The last two solutions are not easily available in Poland, due to there not being an obesity medication registered for the treatment of children. Bariatric surgery can be used in teenagers with morbid obesity meeting the inclusion criteria, those who failed to reach expected results with dietary and behavioural treatment and who have cardiovascular or metabolic complications of obesity [7]. As for following restrictive diets in children, there are more opinions about their disadvantageous effect on growth and development. Pre-made diet products and dietary supplements supporting weight loss are also not recommended [13]. The safe form of prevention and treatment of overweight and obesity in the developmental age is introducing correct nutrition patterns and increasing the physical activity of the whole family [14].

One such pattern is the Pyramid of Healthy Nutrition and Lifestyle of Children and Adolescents, created by experts from the Institute of Food and Nutrition in Warsaw [15]. The Pyramid of Healthy Nutrition is a graphic illustration of the idea of correct nutrition, which is based on Polish nutrition recommendations [16]. Due to the increasing incidence of overweight and obesity in the paediatric population, and still no effective ways of fighting off the spread of those pathologies, a study was created to assess the effectiveness of a one-year dietary intervention in children and adolescents with excess body weight, measured by its effect on chosen anthropometric and biochemical parameters, blood pressure, the incidence of non-alcoholic fatty liver disease and providing nutrition using a simple education tool, which is the Nutrition Pyramid.

Additionally, a correlation was assessed between the nutrition patterns and chosen anthropometric and biochemical parameters, fatty liver and blood pressure in those children before and after the aforementioned intervention.

## 2. Patients and Methods

### 2.1. Choosing the Study Population

The study group consisted of children between the ages of 3 and 18 years old and their legal guardians, hospitalised in the Department of Paediatrics, Endocrinology, Diabetology, Metabolic Diseases and Cardiology of the Developmental Age in the Autonomous Public Clinical Hospital No 1 of Pomeranian Medical University in Szczecin, to be diagnosed for causes of excess body weight.

Taking part in the study was voluntary. Each of the parents and children above the age of 13 were given oral and written information about the aim of the study. Each parent and child above the age of 13 signed an informed consent form to take part in the study. Due to social reasons, participation in the project was proposed for all children with obesity and overweight diagnosed in the Clinic.

The study was permitted by the Pomeranian Medical University Bioethics Committee (decision number KB-0012/34/11 from 16 May 2011).

Children with the following conditions were excluded from the study:Inborn diseases predisposing to obesity, e.g., Down syndrome, Prader–Willi syndrome;Thyroid function disorders;Adrenal gland function disorders;Gonad function disorders;Intellectual disabilities;Chronic diseases, treatment of which may induce an increase in body weight, e.g., steroid therapy.

A total of 94 children aged between 3 and 17 ($\bar{x}$ = 11.4 ± 3.8) with excess body weight were qualified for the study, including 53 (56.3%) girls and 41 (43.6%) boys. Ultimately, 68 (72.3%) children reached the study’s endpoint, aged 4–17 ($\bar{x}$ = 12.4 ± 3.7), with 37 (54.4%) girls and 31 (45.6%) boys.

All children were physically examined by a paediatrician, including puberty assessment using the Tanner scale [17].

### 2.2. The Rules for Anthropometric Measurements, Body Fat Measurement, Fatty Liver Assessment and Blood Pressure Measurement

In children the following anthropometric parameters were measured: body height with an accuracy of 0.1 cm, using a stadiometer (Harpenden type, 602VR, Holtain, Croswell Crymych Pembrokeshire, Wales, UK); body weight with accuracy of 0.01 kg, on a medical scale Radwag WPT 60/150 OW, Radom, Poland; and waist circumference, with an accuracy of 0.5 cm, using a tape measure. Measurements were taken between the costal margins and iliac plates. Hip circumference measurements were taken with an accuracy of 0.5 cm, using a tape measure, along the greater trochanters of the femur.

Results obtained with each measurement were compared to Polish children population norms, established by the OLA and OLAF project [18,19]. Cut-off points defining overweight and obesity in this project are in line the criteria of International Obesity Task Force (IOTF) [5,20].

Based on the obtained measurements body mass index (BMI) was calculated, using the formula: BMI = body weight (kg)/body height (m)^2^. Waist to height ratio (WHtR) was calculated using the formula: WHtR = waist circumference (cm)/body height (cm). Abdominal obesity was diagnosed with waist circumference ≥ 90 percentile for age and sex, and WHtR > 0.5 [21].

To eliminate the effect of age and sex of the examined children on the measured anthropometric parameters, standard deviation scores (SDS) were calculated. To calculate the SDS referential values for the Polish children population from OLA and OLAF project were used [18,19]. In accordance with the presented criteria, overweight was diagnosed in children with BMI ≥ +1SDS and <+2SDS, obesity was diagnosed in children with BMI ≥ +2SDS.

The percentage of adipose and muscle tissue was measured using an electrical bioimpedance analyser (Jawon IOI-353, Selvas Healthcare, Gyeongsan, Korea). The measurement was performed, as per the device producer’s instructions, on fasting children with emptied bladders, in a standing position. Because of the device specifications, measurements were not taken from children below the age of 5.

Blood pressure was measured with a semi-automatic oscillometric apparatus (Omron M1 Compact, Kyoto, Japan). Using 3 separate measurements to obtain an average of systolic and diastolic pressure. Then, standard deviation scores (SDS) were calculated for systolic and diastolic pressure using reference values for the Polish children population from the OLA and OLAF project [22,23]. To assess the frequency of abnormal arterial blood pressure values in the study group, values < 90th percentile for sex, age and height percentile were considered normal. Values between the 90th and 95th percentile were defined as high normal, and values ≥ 95th percentile for sex, age and height percentile were considered as elevated blood pressure.

To screen for the presence of fatty liver, an abdominal ultrasound was performed. However, not all children showed up for the examination.

### 2.3. Biochemical Study Methods

Glucose metabolism was assessed using glucose and insulin levels in 0 and 120 min timepoints of oral glucose tolerance test, after administering 1.75 g/kg glucose (no more than 75 g). Glucose and insulin levels were measured using Cobas C501 device (Roche, Mannheim, Germany).

To assess glucose metabolism guidelines of the Polish Diabetes Association were used [24], setting fasting values as: normal fasting glucose: 70–99 mg/dL; impaired fasting glucose (IFG); 100–125 mg/dL; fasting glucose ≥ 126 mg/dL: diabetes. In 120′ timepoint, the following values were used: normal glucose tolerance (NGT): <140 mg/dL; impaired glucose tolerance (IGT): 140–199 mg/dL; diabetes: ≥200 mg/dL.

Hyperinsulinism in 120′ timepoint of OGTT was diagnosed at insulin levels > 75 µIU/mL [25].

To check the lipid metabolism parameters, the following were measured in the venous blood of children in the study group: total cholesterol, LDL-cholesterol, HDL-cholesterol, triglycerides and alanine aminotransferase, using Cobas C501 device (Roche, Mannheim, Germany).

Lipid metabolism assessment was performed using the guidelines of National Cholesterol Education Program (NCEP) [26], using the following reference values: total cholesterol—<170 (mg/dL); LDL-cholesterol—<110 (mg/dL); HDL-cholesterol—>45 (mg/dL); triglycerides: <75 (mg/dL) (age 0–9); <90 (mg/dL) (age 10–19).

Reference values for ALT: <30 U/L (age 1–12); <23 U/L (age 13–17 girls); <26 U/L (age 13–17 boys).

Using the results, HOMA-IR (Homeostasis Model Assessment—Insulin Resistance) index was calculated, with the formula: HOMA-IR = fasting glucose (mg/dl) × fasting insulin (µIU/mL)/405 [27].

According to available literature, there is no set HOMA-IR value for insulin resistance in children. It has been shown that HOMA-IR increases in puberty [28]. Due to the study group consisting of children of different ages and sex, HOMA-IR percentile values for the Caucasian population were used, with insulin resistance diagnosed in HOMA-IR > 97th percentile for age and sex [27]. Atherogenic index (AI) was calculated using the formula: AI = triglyceride level (mg/dL)/HDL-cholesterol level (mg/dL). All values > 3 were considered incorrect [29].

### 2.4. Diet Analysis

The tool used to assess the patients’ nutrition was a food diary, in which the diet was retrospectively quantitively measured based on 24 h nutrition history (24 h dietary recall), repeated 3 times (2 weekdays, 1 weekend).

The dietary interview was always collected in the presence of the same qualified clinical dietician in a way that ensured each surveyed person understood all the questions. The patient described 3 consecutive days of menus (including 1weekend day) from before hospitalisation. In order to avoid over- or under-reporting of portions eaten by respondents, to estimate portion sizes, the Album of Photographs of Food Products and Dishes by the National Food and Nutrition Institute was used [30]. This album is a collection of photographed products and dishes in 3 different portion sizes converted into grams. The patient, after looking at the photo, indicated the closest size of the product they had used to prepare the meal or the meal itself that they had eaten. Additionally, the participants were encouraged to use the website www.ilewazy.pl, “Photographic weight converter useful in cooking and dieting” [31]. This website is used as a way to recalculate traditional measurement units (such as cup, tablespoon, handful or pinch) into grams. To calculate the calorie and basic macronutrient consumption, DIETA.5 program by the National Food and Nutrition Institute was used [32].

Caloric demand was calculated for each child individually and compared to Polish population nutrition norms [16]. Eliminating the age, sex and physical activity levels of the study group, the percentage was calculated of the norms for energy, protein, total fat, saturated fat, cholesterol, carbohydrates, “simple sugars” and dietary fibre, individually for each child. Based on a survey study, the amount of physical activity was calculated. The following activity levels were described: low—to 6 h/w, including physical education classes; intermediate—7–10 h/w, including physical activity classes; high—≥11 h/w, including physical education classes and extracurricular sport activities.

To set the recommendations of macronutrient consumption, the percentage of total caloric demand was established: proteins 12.5%, carbohydrates 57.5%, “simple sugars” (saccharose and lactose) 15.0%, total fats 30.0%, and saturated fats 10.0%.

The recommended consumption of dietary fibre was established as adequate intake (AI) [16], and correct cholesterol consumption level, as per nutritional prophylactic guidelines, as 300 mg/day [33].

The patients’ nutrition was assessed and graded by calculating the consumption of energy and macronutrients in the diet in comparison to set recommendations, as percentages of said values.

For nutritional education “Pyramid of Healthy Nutrition for Children and Adolescents” was used, along with accompanying it 10 Rules of Healthy Nutrition of Children and Adolescents, created by the National Food and Nutrition Institute in Warsaw, 2009 [16].

During the study and education process, particular emphasis was put on: decreasing saturated fat consumption, by eliminating calorie-dense snacks, such as potato chips, salty sticks and fast foods; eating more steamed, boiled and stewed meals in place of fried ones; increasing consumption of fish and lean meat and dairy; increasing the amount of consumed vegetables to three main meals and fruit to two meals; decreasing the consumption of sugar, candy, sweet yoghurts and dairy desserts; eliminating sweetened beverages and fruit juices introducing whole grain cereal products; daily water consumption; regular consumption of a proper number of meals; increasing control over nutritional behaviours and informed decisions considering choosing the right product—the ability to analyse the products’ labels; and encouraging daily physical activity.

### 2.5. Organisation of Dietary Education

Educational measures were aimed at both children and their parents or carers during all visits. During the study, children took part in 6 individual education meetings. During the first 3 months, they were controlled once every month, control 4–6 took place once every 3 months. For the 6th visit, children along with their legal guardians were invited back to the Department to perform control laboratory tests, anthropometric measurements and a nutrition assessment. On every visit, the children and their guardians were re-educated. During the education special emphasis was placed on: reducing the intake of saturated fats in the diet, by: eliminating high-calorie snacks such as crisps, nuts, salty sticks and fast-foods; limiting fried meals; increasing the consumption of fish, lean meat and dairy products; increasing the amount of fruit and vegetables consumed; limiting the consumption of sugar, sweets, sweetened dairy products; eliminating sweetened drinks and fruit juices; introducing whole grain cereal products; drinking water daily; eating an adequate amount of food regularly; making conscious decisions to choose healthy foods—being able to analyse product labels; and encouraging daily physical activity. Children and parents/carers were encouraged to eat together at the table and avoid eating meals and snacks in front of the TV, computer or smartphone. The current nutrition diary was analysed, and all the errors were corrected. Easy, cheap and healthy ways of reducing the caloric value of consumed meals were proposed. Children and their parents were motivated to together change the incorrect nutrition habits. During the follow-up meetings, special attention was paid to the opportunity to ask questions and to ensure that sufficient time was given for each visit (40–60 min on average). The study presented the children’s parents with an opportunity for one-on-one meetings with an experienced dietitian, giving greater opportunity to focus on the child’s personal problems, preferences and dietary predispositions. Patients also had the opportunity to send questions to the educator via e-mail and telephone text messages. Most importantly, respondents also had the opportunity to adjust their individual appointment from among the 3 proposed dates, thus avoiding missed appointments.

### 2.6. Statistical Methods

To assess the effect of dietary intervention on quantitative parameters, for every child, the difference was calculated between the value of the measured parameter after and before the intervention. The significance of said differences was analysed using Wilcoxon signed rank test. To compare qualitative parameters measured before and after the intervention, McNemar’s χ^2^ test was used. The strength of correlations between quantitative parameters was assessed using the value of Spearman rank correlation coefficient (r_s_). As a statistical significance threshold, a value of *p* < 0.05 was chosen. All calculations were performed using Statistica 13 program (StatSoft, Kraków, Poland).

## 3. Results

### 3.1. Study Group Characteristics

Of the 94 children qualified for the study, 26 (27.7%) resigned from further education, at different stages of the study. Children who did not reach the endpoint of the study came from families living in rural areas, and their mothers mostly had only primary or vocational education.

In terms of anthropometric parameters, children who completed the dietary intervention did not differ from children who dropped out of the intervention. The data are presented in Table 1.

The education level of the studied children’s mothers varied. Almost half (47.8%) of them had secondary education, more than a third (35.8%) had higher education, and primary or vocational education, respectively, 8.9% and 7.5%. The vast majority of mothers (83.6%) were professionally active. In the study group, two-thirds (64.2%) of the mothers lived in urban areas, and one-third (35.8%) in rural areas. Nearly three-quarters (68.7%) of the mothers were also characterised by excessive body weight. It was observed that those who dropped out of the study were characterised by a “lower degree” of motivation to change their children’s eating habits and reduce their weight, compared to those who remained in the study until its completion. This may suggest that the patients who remained in the study were characterised by a higher-than-average degree of motivation to lose weight, which translated into their weight reduction outcomes. The need for the children and their parents to dedicate a huge amount of time during frequent follow-ups over the 12-month dietary intervention period was also an obstacle.

A total of 68 children underwent further analyses. Based on BMI SDS, 59 (86.8%) were diagnosed with obesity and 9 (13.2%) with overweight. In 41 (60.3%) children, BMI exceeded +3SDS, and in almost one in ten children, +6 SDS. Because of a relatively small study group, children with overweight and obesity were analysed together.

### 3.2. The Effect of One-Year Dietary Intervention on Anthropometric Parameters

One-year dietary education contributed to a significant (*p* < 0.001) improvement in body weight, waist and hip circumference and WHtR index (Table 2). A significant increase in lean muscle mass and decrease in adipose tissue percentage in bodies of examined children were also observed.

### 3.3. Effect of Dietary Intervention on Lipid Profile Parameters, Fatty Liver and Blood Pressure

In the study group, after the 12-month intervention, an improvement was observed in almost all lipid profile parameters, except for total cholesterol levels (Table 3). There was a significant increase in HDL-cholesterol (*p* < 0.001) and significant decrease in ALT (*p* = 0.0001) and TG/HDL-CH ratio (*p* = 0.0009). Values of systolic and diastolic blood pressure presented as SDS showed no significant differences before and after the intervention.

The assessment of the effect of the dietary intervention on the presence of fatty liver was performed on 50 children. Before the intervention fatty liver was diagnosed in 26 (52.0%) of children, and after the intervention in 12 (24.0%) of them. The one-year intervention contributed to the resolution of this pathology in 16 (32.0%) children. In 2 children (4.0%), fatty liver appeared after the dietary education. A significant 28.0% decrease in the incidence of fatty liver in examined children was observed (*p* = 0.002).

### 3.4. The Effect of One-Year Education on Chosen Parameters of Glucose Metabolism

As seen in Table 4, a significant effect of dietary education on the improvement in all measured glucose metabolism parameters was shown.

### 3.5. The Effect of Dietary Education on Consumption of Energy and Macronutrients

The one-year intervention improved the nutritional habits of children in the study group (Table 5). The amount of consumed energy significantly changed (*p* < 0.001), as did the makeup of macronutrients and the realization of assumed norms for the considered macronutrients, except for dietary fibre, in which the effect was bordering on statistical significance. Significant changes were seen in the percentage of energy coming from proteins (*p* = 0.0002), saturated fat (*p* = 0.01) and “simple sugars” (*p* = 0.0009). For carbohydrates and total fats, the result was bordering on statistical significance.

Before the dietary intervention, children’s portions were incorrectly balanced. The percentage of realised norms for energy and main macronutrients and their percentage share in the energy value of the diet was exceeded in almost all analysed macronutrients, except for carbohydrates. Despite that, the amount of consumed carbohydrates significantly decreased in absolute values (*p* < 0.001).

The caloric value of the children’s diet, both before and after the intervention, varied widely, as confirmed by high values of standard deviations from mean values. Before the intervention, the consumption of energy in three-quarters of the respondents was higher by about 30.0% of their caloric needs. After the intervention, the consumption of energy was significantly reduced (*p* < 0.001)

After the dietary intervention, the amount of consumed protein was also significantly reduced (*p* = 0.006). Its percentage share of the total caloric value of the diet was significantly increased though (*p* = 0.0002), reaching a value of 17.35 ± 3.25%.

The consumption of energy coming from fats was on average 32.23 ± 6.25%, with a recommended 30.0%. After a one-year dietary intervention, the percentage of energy coming from fats changed, reaching the value of 30.32 ± 6.95%. This change bordered on statistical significance. The consumption of total fats was significantly decreased, down to 83.10 ± 29.40% of the norm realization (*p* < 0.001).

The biggest reduction was seen in the consumption of saturated fats and “simple sugars”. Before the change in diet, the consumption of saturated fats exceeded the recommended norm, on average by 144.07 ± 52.59%. After one year of education, the amount of energy from saturated fat consumption significantly decreased (*p* = 0.01) and was on average 100.28 ± 40.91% of the norm. After the dietary intervention, the consumption of “simple sugars” was reduced almost by half, from 112.36 ± 68.85% to 67.80 ± 43.40% of the recommended norm. This change was statistically significant (*p* < 0.001).

The consumption of cholesterol exceeded 300 mg/day, on average 347.99 ± 127.16 mg/day. After the dietary intervention, the amount of consumed cholesterol was significantly reduced (*p* = 0.0001), on average to 269.75 ± 131.93 mg/day.

The amount of dietary fibre consumed was not reduced. It was comparable to values from before the dietary intervention.

### 3.6. Correlation between Changes in Macronutrient Consumption and Chosen Somatic Development Parameters and Blood Pressure in Children after Dietary Intervention

The study showed only singular correlations between macronutrients and measured anthropometric parameters in children with excess body weight (Table 6).

After the one-year dietary intervention, a statistically significant positive correlation was observed between the consumption of protein and WHtR (r_s_ = 0.25, *p* = 0.03), as well as between carbohydrate consumption and WHtR (r_s_ = 0.26, *p* = 0.03) and the percentage of adipose tissue in the body (r_s_ = 0.27 *p* = 0.03).

Analysis of macronutrient consumption and measured parameters of lipid profile is shown in Table 7. As shown in the table, only singular correlations between macronutrient consumption and lipid profile parameters were observed.

After the dietary intervention, a statistically significant positive correlation was shown between protein consumption and LDL-cholesterol levels (r_s_ = 0.28, *p* = 0.02). A significant, positive correlation was also shown between fat consumption and triglyceride levels (r_s_ = 0.26, *p* = 0.03).

There was no correlation between other macronutrients and measured parameters of lipid profile.

Analysis of macronutrient consumption and the measured parameters of glucose metabolism after the dietary intervention is shown in Table 8. A significant, positive correlation was observed between carbohydrate consumption and fasting glucose (r_s_ = 0.24, *p* = 0.01) and insulin at 120′ timepoint of OGTT (r_s_ = 0.35, *p* = 0.04). There was also a positive correlation between the consumption of “simple sugars” and glucose levels in 120′ of OGTT (r_s_ = 0.25, *p* = 0.04) and insulin levels at 120′ of OGTT (r_s_ = 0.32, *p* = 0.01).

No significant correlations were observed between other analysed macronutrients and measured glucose metabolism parameters.

After the dietary intervention, no significant correlation was shown between macronutrient consumption and arterial blood pressure in children with excess body weight participating in the study.

## 4. Discussion

The coincidence of excess body weight and its metabolic complications is treated as a high-risk situation, requiring an intervention in lifestyle, of which an important element, along with physical activity, is a change in dietary habits [34]. A change in nutrition is not only a sage and non-invasive method of weight reduction, but also sometimes is the only way to improve the biochemical and anthropometric parameters in the developmental age population. Sisson et al. [35] showed that, while performing a systematic analysis of interventions in obese children, out of 45 dietary interventions, 87.0% led to expected results. Dietary interventions are more effective when parents are involved in their realisation [36]. Such interventions are called “the golden standard of fighting obesity in children” [37]. Furthermore, Pakpour et al. [38] showed that engaging the parents has a significantly bigger effect on the improvement of biochemical and anthropometric parameters as well as decreasing the caloric intake than dietary intervention alone.

This dietary intervention was performed with the participation of one of the parents. It was based on the guidelines of healthy nutrition and the Pyramid of Nutrition of Children and Adolescents from 2009 [16]. The intervention showed a significant effect on improving both parameters of somatic development and biochemical parameters in children with excess body weight. As a result of a yearlong intervention, the following changes in somatic development parameters were observed: SDS BMI −0.8; waist circumference SDS −0.77; WHtR −0.04; adipose tissue weight −1.76 kg; adipose tissue percentage −2.14; lean muscle mass +1.34 kg.

A different study showed a significantly less pronounced therapeutic effect on the reduction in BMI SDS and waist circumference SDS, with similar values of adipose tissue percentage reduction [39]. In the WATCH IT study, there was an increase noticed both in BMI SDS and adipose tissue percentage of 0.03SDS and 1.4%, respectively. Said result was explained to be an effect of, among other things, the parents not being involved in the dietary education of children taking part in the study [40]. In studies performed by Savoye et al. [41], after a 12-month dietary intervention, a reduction in adipose tissue mass of 3.7 kg was observed. In this study, a higher success rate was achieved by engaging the families in active physical exercises. It is known that moderate physical activity alone does not cause body weight loss, but together with caloric restriction it helps achieve and sustain significant weight loss [42]. Therefore, one of the elements of education in our study was encouraging children and their parents to engage in additional physical activity. However, after the end of the intervention, no increase in the declared time of physical activity during the week was observed. This means that in our study the participants reached a significant reduction in measured anthropometric parameters thanks only to personalised dietary intervention, with the parent’s engagement. Another factor possibly affecting greater body weight reduction could be the patients’ age. The GECKO study [43] showed that weight loss and decreasing waist and hip circumference in younger children is more effective and seems easier to achieve. One of the reasons for this could be a shorter period of obesity in those children. In the current study, both in girls and boys, one-fifth of the study group were children below the age of 7. Additionally, many authors note greater weight loss in children with severe obesity (BMI > 3 SD) [44]. In our study, the mean BMI SDS was +3.7. Moreover, in body composition analysis, along with a reduction in adipose tissue weight, an increase in lean muscle weight was observed. Those results seem significant, as, in some studies, the decrease in adipose tissue weight was accompanied by a reduction in lean muscle weight [45]. The decrease in lean muscle weight causes “decreased metabolic rate”, and as a consequence, “slower weight loss” [46]. Because of this, it can be assumed that participants in our study will not regain the “lost” weight.

In this study, a correlation between the consumption of macronutrients and its effect on the measured anthropometric parameters was analysed. A significant, albeit weak, positive correlation was found between the amount of consumed protein and carbohydrates and WHtR, as well as between the amount of consumed carbohydrates and the adipose tissue percentage in the body.

In the studies by Starbała et al. [47], similar correlations were described between BMI SDS and the percentage of carbohydrates and fats in the diet. A different study of Italian children with overweight and obesity in the age of 9–13, assessing the energy consumption and diet composition, showed a negative correlation between fat and carbohydrate consumption and WHtR, and a positive correlation with the consumption of proteins [48]. In a study by Walker et al. [49], in a group of children aged between 7–15, it was shown that using a low-calorie diet with elements of low glycaemic index significantly affected the reduction in WHtR and adipose tissue percentage in the body.

In our study, a rather high percentage of adipose tissue was observed in the examined children (32.27 ± 5.8%), which was localised centrally in all patients (WHtR = 0.60 ± 0.06). Factors responsible for excess body fat and its central distribution were improperly balanced diet and incorrect eating habits. Considering WHtR is a strong risk factor for circulatory diseases and metabolic syndrome [50], and can be considered a significant marker in clinical practice [51], it may be considered useful to know that not only the amount of consumed carbohydrates has a deciding effect on the development of abdominal obesity [52], but also the amount of consumed protein. Proteins are considered a “safe’’ macronutrient in the developmental age in the amount not exceeding 10.0−20.0% of the diet’s energy value [16]. In younger children, the amount of protein does not have restrictive upper values, as there is no convincing evidence for detrimental effects of excess consumption of this macronutrient in children above the age of 1 [53]. Additionally, protein is the only macronutrient not modified when constructing low-calorie diets, whereas the amount of carbohydrates and fat does become modified. Considering the consumption of protein can also modulate the amount of adipose tissue in the body [54], there is a need for further studies aiming to find an optimal amount of protein to be consumed by children with excess body weight. Known recommendations apply to healthy children [16], and there is a lot of controversy surrounding its calculation to current or correct body weight. In this study, the amount of protein consumed during the yearlong intervention significantly decreased. Considering the lower caloric intake, however, its percentage in the diet significantly increased.

Goluch-Koniuszy and Fugiel [55], assessing the nutritional status and dietary habits of teenage girls, found that almost 1 in 10 15-year-olds and 1 in 4 16-year-olds present with visceral obesity, with energy consumption from protein at 14.4% and 14.1%, respectively. Agostoni et al. [56] suggested that a correlation between higher protein consumption and later obesity is found mostly in populations with protein exceeding 15.0–16.0% of total energy consumption.

In studies on adults, a positive effect of a high-protein diet on body weight reduction was shown, due to increased satiety and higher energy expenditure caused by more effective thermogenesis after a high-protein meal. Unfortunately, it is still unknown if a high-protein diet shows similar mechanisms leading to a decrease in body weight and improving satiety in children with excess body weight [57]. It seems, thus, that in the case of children, the effect of protein consumption on body fat cannot merely be measured in absolute values. After changing nutritional habits, along with a reduction in macronutrients, the caloric value of daily provisions is also reduced. Some of the macronutrients can then see an increase when considered as a percentage of energy consumption.

Changes in the anthropometric parameters observed in our study were connected to changes in measured biochemical parameters. A one-year dietary intervention significantly improved the lipid profile, reduced liver adiposity and improved glucose metabolism parameters. There was no observed effect of the dietary intervention on systolic and diastolic blood pressure. There was also no correlation between blood pressure and macronutrient consumption, despite positive changes in dietary habits and reduced calorie consumption. It seems that the factor having a greater effect on blood pressure is physical activity. It was confirmed by Vasconcellos et al. [58], who performed a prospective review of the literature concerning the effect of physical activity on circulatory system diseases in children with overweight and obesity.

The correlation between obesity and incorrect lipid profile [7,59,60], non-alcoholic fatty liver disease [61,62] and glucose metabolism disorders [43,63] has been confirmed in numerous studies.

In this study, our one-year dietary education caused a significant decrease in levels of LDL-cholesterol (on average 6.29 mg/dL), triglycerides (16.95 mg/dL) and activity of alanine aminotransferase (7.19 U/L). The patients significantly “improved” the levels of HDL-cholesterol (5.31 mg/dL), which resulted in a significant decrease in the atherogenic index, i.e., the ratio of triglycerides to HDL-cholesterol (by 0.65). Similar effects were observed in other studies [46,64,65], while other ones did not achieve such results [66,67,68].

One of the factors that could play a role in the improvement of lipid profile is a change in BMI SDS. The correlation between BMI SDS reduction, circulatory system health and body composition has been analysed in many studies [69,70]. It was observed that a change of 0.25 in BMI SDS can be considered clinically significant for the improvement in fasting insulin sensitivity and proportion of total cholesterol/HDL-cholesterol. Bigger improvements were observed with a BMI SDS reduction of 0.5 [71]. In our study, after a one-year dietary intervention, a change of −0.80 in BMI SDS was observed.

As is known, the improvement of the lipid profile does not only depend on body weight and adipose tissue percentage reduction, but also on the diet composition [33]. In our study, a correlation was assessed between the macronutrient and metabolic status of the body. A positive correlation was found between protein consumption and LDL-cholesterol, as well as fat consumption and triglyceride levels. Similarly, Reinehr et al. [72] showed that a decreased consumption of fats and energy leads to an improved atherogenic profile. However, Sung et al. [73] showed that a low-calorie diet with fat restriction does not always lead to the aforementioned effect, because such dietary models lead to increased carbohydrate consumption, which, in turn, causes an increase in triglyceride levels. It seems then that a change in the proportions and quality of macronutrients alone can improve the lipid profile and lead to intended changes, without changing the amount of consumed energy [74]. Such a therapeutic intervention is safe in children with overweight and obesity. Taking into consideration the significance of protein and fat consumption’s effect on the lipid profile, more attention should be given to the amount of these macronutrients as prophylaxis of abdominal obesity in the paediatric population.

A surprising finding in our study is a lack of a significant correlation between the amount of consumed carbohydrates and lipid profile parameters. A lack of such a correlation has also been described in other publications [75]. It can be explained by the fact that before our intervention, children already consumed relatively few carbohydrates, with only a small excess of “simple sugars”, which was reduced after the one-year dietary intervention.

It is believed that the excess consumption of carbohydrates, especially from sweetened beverages, increases the risk of non-alcoholic fatty liver disease. The risk is increased more than twofold when consuming refined sugars in beverages, as found in a group of children with obesity in the age group of 3–14 [76]. In the proprietary study, there was no correlation proven between the dietary habits and non-alcoholic fatty liver disease, and none of the analysed macronutrients were especially responsible for this complication. This can be a result of a small study group that was statistically analysed. It can also be assumed that other, not strictly dietary factors, play a greater role in the development of NAFLD, such as insulin resistance [77].

In the available literature, there are not many papers assessing the consumption of macronutrients in children with obesity and the effect of dietary interventions on its change. There are ongoing efforts to improve and verify the techniques of collecting dietary data from subjective examination of the patient, as well as methods of assessing the nutritional habits. One of those methods is a 24 h nutrition diary [78]. In this study, such a tool was used, which allowed us to perform a thorough analysis.

Before the intervention, the children’s meals were improperly balanced, and the realisation of the suggested norms of energy and macronutrient consumption, as well as macronutrients consumption as a percentage of the whole energy intake, was exceeded for almost all analysed values, except for carbohydrates. Considering the study had no control group consisting of healthy children with correct diets, one has to be especially careful when formulating any conclusions. After the dietary intervention, participants improved the results concerning the quality of their diets and came closer to the recommended nutritional norms, mostly by achieving values closer to recommendations with regard to saturated fats and dietary fibre. A similar observation was made by Verduci et al. [79]. After a one-year intervention, in 90.0% of children with obesity in ages 6–15, after implementing a diet well balanced with regard to macronutrient distribution, they achieved a significant reduction in calorie consumption (604.84 kcal) and redistribution of macronutrients in the recommended range. In another study of children and adolescents with abdominal obesity, after an interdisciplinary intervention, energy consumption was reduced by on average 313.1 kcal, carbohydrates by 71.7 g, protein by 11.1 g, fat by 44.4 g, saturated fat by 15.3 g and cholesterol by 78 mg. Despite these changes, the consumption of carbohydrates and fat still exceeded the norm [80]. In our study, a significant decrease in consumed energy was achieved (on average by 461.8 kcal/day), which resulted in a decrease in the consumption of carbohydrates (by 72.14 g), fat (19.83 g), saturated fat (9.09 g), cholesterol (78.24 mg) and protein (8.10 g). The amount of protein in the diet, despite its reduction, still exceeded the normative value. 

In summary, when analysing the diet, it was noted that the consumption of protein decreased significantly in absolute values, but its value as a percentage of total energy intake was increased, with a simultaneous decrease in fat consumption to 30.0%, and carbohydrates to less than 50.0% of total energy intake. Children in the study also reduced the amount of consumed energy, down to 83.0% of the suggested norm. Such results may suggest that improving nutrition habits, without planning for a decrease in energy consumption, may reduce caloric intake. This resulted in a significant reduction in body weight, which, in turn, affected the decrease in measured lipid profile parameters. The change in energy consumption coming from protein, fats and carbohydrates affected the correlations described in the study. Children whose protein consumption decreased had a lower WHtR, decreased LDL-cholesterol levels and fasting glucose, and a lower HOMA-IR. In turn, children whose fat consumption decreased, had lower triglycerides. The biggest changes, both in anthropometric measurements and biochemical studies, were seen in children whose carbohydrate consumption was reduced. Those children presented lower WHtR and a bigger reduction in adipose tissue percentage, lower fasting glucose levels and insulin in the 120′ timepoint of OGTT. Additionally, the lower consumption of “simple sugars” could cause a significant improvement in insulin sensitivity and β-cell function, which plays a major role in lowering the glucose levels in the 120′ timepoint of OGTT. Therefore, the WHO recommendations published in 2015 suggesting reducing simple sugar consumption below 10.0% of total daily caloric intake (with 5.0% being a preferred recommendation) to reduce the risk of development of obesity and type 2 diabetes mellitus seem justified [81]. In our study, the consumption of simple sugars after the intervention was less than 12.0% of total daily caloric intake.

The reduction in risk factors of metabolic complications of excess body weight in studied children has significant clinical implications. Incorrect lipid profile, glucose metabolism disorders, fatty liver and hypertension—are all determinants of the higher mortality of people with obesity. The improvement in lipid and glucose metabolism in our study is thus as significant as the one achievable with pharmacological treatment [82], without the possibility of adverse effects of used medication.

Despite this, modifying individual behaviours still remains a milestone in obesity prophylaxis in the paediatric population. However, regardless of the way that nutrition is modified, and the tools and time planned for the intervention, in the case of children, it is crucial that parents take part in it as well [83].

In our study, an opportunity was created for individual meetings of one of the parents with an experienced dietitian. Making use of individual meetings allows the focus to be on personal problems, preferences and predispositions regarding nutrition. Adjusting the way of giving out information to the receptive ability of a single listener may also be more effective than group counselling [84].

Our study had some limitations. The lack of a control group made it impossible to compare the effect and strength of the intervention. All children, both with excess body weight and normal body weight and their families, should have the same opportunities when it comes to finding information about correct nutrition rules and current recommendations and guidelines. Another limitation comes from the size of the population that underwent the intervention. The intervention was aimed only at children who were referred to the Department to diagnose for causes of obesity. This group may not be fully representative of all children with obesity. Hospitalised children, by actively looking for ways of treatment, are more motivated to reduce their body weight. Additionally, the number of children in the study group was restricted due to the time-consuming nature of individual education. During the 12-month intervention, 26 (27.7%) children gave up on participating in the study at different stages of its duration. The amount of time required for the patients to show up for frequent checkups during the 12-month dietary intervention was also an obstacle. It must be noted that results in our study group were not divided according to children’s age and sex, which is another limitation caused by the small sample size in our study. What also needs to be taken into account is the way the surveys were filled and the possibility of under- or overestimating the consumed foods, which may be significant in the prophylaxis and treatment of excess body weight. Additionally, total energy intake may not accurately represent the actual energy intake from meals prepared at home. The child’s diet is affected by many factors, including their peers. Underestimation of food intake is an important obstacle in the studies of obesity. Children with excess body weight tend to underestimate the amount of food they are consuming, therefore sometimes using the nutrition diary does not fully work, because not every dietary error is noted there. The study used a 24-h dietary interview, which is considered one of the more accurate methods of estimating average intake with a questionnaire. It is characterised by a lower possibility of error concerning the omission or addition of a product or dish. This method also has the advantages of a relatively short time between food consumption and the recall thereof, and a fairly high accuracy of the data obtained. Because of new food items appearing constantly and the lack of comparable items, the existing database to calculate the caloric value and macronutrients may vary from actual values. The precise control and assessment of a strictly defined diet and its effect on biochemical parameters in children are only possible in hospitals. However, the results gathered during that time are able to assess the “real” therapeutic possibilities in everyday life. Additionally, the connection between energy consumption and energy balance must be interpreted in the context of energy expenditure, an important part of which is physical activity. Physical activity is difficult to verify by a dietitian. Its self-reporting is also biased. Because of that, in our study, no attempt was made to assess the effect of physical activity on biochemical parameters and the state of nutrition in children. Despite the aforementioned limitations, it was shown that education directed at changing the whole family’s dietary habits, using a simple model of the Nutrition Pyramid [15], allowed most of the children to modify their nutrition. It also led to an improvement in measured anthropometric and biochemical parameters, without the use of pharmacotherapy. It was also shown that the methodology used by us in this study can be as effective as that described in literature multi-specialist interventions, using nutritional models such as the Mediterranean diet, DASH (Dietary Approaches to Stop Hypertension), low GI or limiting calories. Considering that prophylactic diets and diets for the treatment of cardiovascular complications in children have not yet been unequivocally defined, our observations can be helpful in their creation. More studies are required though concerning the cause-and-effect assessment of the undertaken interventions. 

## 5. Conclusions

In the studied group of children, a one-year intervention contributed to a significant reduction in body weight, waist and hip circumference and adipose tissue percentage in the bodies of those children. In part, the patients’ resolution of the symptoms of fatty liver was observed. Moreover, a significant improvement in measured lipid profile parameters was observed, except for total cholesterol. There was also no effect of the dietary intervention on arterial blood pressure.The one-year dietary intervention also significantly changed the amount of consumed energy and macronutrients, except for dietary fibre. The biggest reduction was observed in saturated fat and simple sugars, i.e., dietary components, which when consumed in abundance are considered one of the main causes of metabolic complications of obesity.The reduction in consumption of protein and carbohydrates significantly affected the reduction in the adipose tissue percentage of the examined children, decreasing the risk of excess body weight complications. It seems justified to educate parents on the nutritional value of products being a source of protein, because of the presence of fats and carbohydrates in these products.A one-year dietary education of children with excess body weight has important clinical significance, both in body weight reduction and limiting the possible complications of overweight and obesity. It seems then justified to provide easier access to qualified healthcare personnel, i.e., not only doctors, but also dietitians.

## Figures and Tables

**Table 1 ijerph-19-11686-t001:** Comparison of anthropometric parameters of children who completed the dietary intervention with those who did not.

Measured Parameter	Finished	Did Not Finish	*n* Finished	*n* Did Not Finish	*p*-Value
	$\bar{x}$ ± SD	$\bar{x}$ ± SD			
Age	11.35 ± 3.63	11.43 ± 4.16	68	26	0.97
Weight SDS	3.47 ± 1.57	3.84 ± 1.95	68	26	0.27
Height SDS	0.87 ± 1.11	0.48 ± 0.95	68	26	0.058
BMI SDS	3.70 ± 1.68	4.36 ± 2.00	68	26	0.13
Waist circumference SDS	4.27 ± 1.79	4.85 ± 2.23	68	25	0.27
Hip circumference SDS	2.85 ± 1.37	3.52 ± 1.73	68	25	0.09
WHtR	0.60 ± 0.06	0.62 ± 0.08	68	25	0.15

BMI—body mass index; WHtR—waist/height ratio; SDS—standard deviation score; $\bar{x}$—mean value; *p*—probability.

**Table 2 ijerph-19-11686-t002:** Changes in measured anthropometric parameters before and after the dietary intervention.

Measured Parameter	*n*	Before	After	Difference in Measured Parameters	*p*-Value
		$\bar{x}$ ± SD	$\bar{x}$ ± SD	$\bar{x}$ ± SD	
Weight SDS	68	3.47 ± 1.57	2.83 ± 1.62	−0.64 ± 0.82	<0.001
BMI SDS	68	3.70 ± 1.68	2.90 ± 1.70	−0.80 ± 0.96	<0.001
Waist circumference SDS	68	4.27 ± 1.79	3.50 ± 1.92	−0.77 ± 1.23	<0.001
Hip circumference SDS	68	2.85 ± 1.37	2.07 ± 1.48	−0.78 ± 0.97	<0.001
WHtR	68	0.60 ± 0.06	0.56 ± 0.07	−0.04 ± 0.04	<0.001
Adipose tissue (kg)	66 *	24.62 ± 12.56	22.86 ± 12.23	−1.76 ± 4.85	0.03
Adipose tissue (%)	66 *	32.27 ± 5.8	30.13 ± 7.02	−2.14 ± 3.54	<0.001
Lean muscle weight (kg)	66 *	43.89 ± 4.47	45.23 ± 14.01	1.34 ± 2.83	<0.001

BMI—body mass index; WHtR—waist/height ratio; SDS—standard deviation score; $\bar{x}$—mean value; *—in 2 children adipose tissue and muscle weight were not measured due to young age (<5).

**Table 3 ijerph-19-11686-t003:** Changes in lipid profile parameters and blood pressure before and after the dietary intervention.

Measured Parameter	*n*	Before	After	Difference in Measured Parameters	*p*-Value
		$\bar{x}$ ± SD	$\bar{x}$ ± SD	$\bar{x}$ ± SD	
Total cholesterol (mg/dL)	67	166.11 ± 27.62	161.91 ± 29.9	−4.20 ± 20.39	0.22
HDL-CH (mg/dL)	67	46.19 ± 10.98	51.5 ± 13.8	5.31 ± 7.72	<0.001
TG (mg/dL)	67	114.45 ± 60.59	97.5 ± 49.4	−16.95 ± 53.98	0.02
LDL-CH (mg/dL)	66	108.89 ± 26.12	102.6 ± 29.4	−6.29 ± 19.23	0.02
TG/HDL-CH ratio	67	2.75 ± 1.97	2.1 ± 1.4	−0.65 ± 1.58	0.0009
ALT (U/L)	61	26.29 ± 17.84	19.1 ± 9.3	−7.19 ± 15.78	0.0001
Systolic pressure SDS Diastolic pressure SDS	68 68	1.18 ± 1.03 1.14 ± 0.82	1.03 ± 1.01 1.24 ± 0.91	−0.15 ± 1.11 0.10 ± 0.91	0.11 0.53

HDL-CH—HDL-cholesterol; TG—triglycerides; LDL-CH—LDL-cholesterol; ALT—alanine aminotransferase; SD—standard deviation; SDS—standard deviation score; $\bar{x}$—mean value; *n*—number of children.

**Table 4 ijerph-19-11686-t004:** Measured glucose metabolism parameters before and after the dietary intervention.

Measured Parameter	*n*	Measured Parameter	After	Difference in Measured Parameters	*p*-Value
		$\bar{x}$ ± SD	$\bar{x}$ ± SD	$\bar{x}$ ± SD	
Fasting glucose (mg/dL)	67	88.44 ± 7.87	86.25 ± 8.58	−2.19 ± 9.39	0.04
Glucose at 120′of OGTT (mg/dL)	65	110.90 ± 20.32	102.80 ± 17.67	−8.10 ± 27.43	0.01
Fasting insulin (µIU/mL)	66	21.37 ± 10.86	16.46 ± 8.35	−4.91 ± 11.56	0.001
Insulin at 120′of OGTT (µIU/mL)	64	125.02 ± 105.09	86.42 ± 68.19	−38.60 ± 98.08	0.005
HOMA-IR	65	4.74 ± 2.49	3.53 ± 1.87	−1.21 ± 2.72	0.002

OGTT—oral glucose tolerance test; $\bar{x}$—mean value; SD—standard deviation; HOMA-IR—insulin resistance index; *n*—number of children.

**Table 5 ijerph-19-11686-t005:** The effect of one-year dietary education on the consumption of energy and chosen macronutrients by children in the study group.

Measured Parameter		*n*	Before Dietary Intervention			After Dietary Intervention			Difference	*p*-Value
			Q_1_	Me (Min.–Max.)	Q_3_	Q_1_	Me (Min.–Max.)	Q_3_	Me Q_1_ ÷ Q_3_	
Energy	(kcal/day)	68	1824.5	2183.45 (1331.80–4719.20)	2595.5	1472.4	1741.30 (845.30–3500.00)	2123	−412.8 −750.5 ÷ 18.20	<0.001
	(%) norm *		81.5	103.36 (55.57–197.38)	129.8	61.1	85.20 (30.30–170.98)	97.7	−23.7 −32.7 ÷ 53.8	<0.001
Proteins	(g/day)	68	70.7	79.95 (38.55–148.25)	100.1	61.9	75.05 (26.00–175.00)	88.8	−7.9 −22.8 ÷ 7.0	0.006
	(%) norm *		98.8	129.86 (50.02–282.74)	160.1	81.4	109.92 (39.61–215.38)	133.9	−17.3 −38.5 ÷ 4.4	0.0003
	% E from P		13.3	14.91 (9.25–26.43)	17.1	15.7	17.39 (6.11–26.20)	18.7	2.25 −0.17 ÷ 4.41	0.0002
Fat	(g/day)	68	58.7	74.30 (34.70–171.80)	96	41.4	58.75 (17.50–139.60)	78.7	−14.2 −38.1 ÷ 1.7	<0.001
	(%) norm *		83.9	109.71 (45.87–228.00)	146.4	59.4	82.12 (25.00–151.58)	109.8	−23.2 −62.8 ÷ −0.9	<0.001
	% E from F		27.4	32.87 (18.44–47.92)	47.9	25.6	30.35 (9.26–46.47)	35.1	−6.9 −18.8 ÷ −0.3	0.052
Saturated fats	(g/day)	68	23.7	32.10 (12.80–70.00)	42.1	16.2	22.95 (8.90–64.00)	29.9	−6.15 −19 ÷ −1.2	<0.001
	(%) norm *		105.7	140.78 (53.58–272.57)	174.7	72.1	97.60 (31.15–221.53)	122.3	−33.42 −75.3 ÷ −10.4	<0.001
	E from SF		10.7	13.79 (6.80–25.55)	15.7	9.7	11.75 (4.71–20.50)	14.1	−1.7 −3.3 ÷ 1.6	0.01
Cholesterol	(mg/day)	68	266.6	342.20 (62.00–698.50)	452.9	192.5	239.00 (33.40–751.30)	331.6	−79.7 −203.2 ÷ 7.2	0.0001
	(%) norm *		82.2	114.06 (20.66–232.83)	150.9	64.2	79.66 (11.13–250.43)	110.5	−26.6 −67.7 ÷ 2.4	0.0001
Carbohydrates	(g/day)	68	236.1	289.70 (155.10–780.10)	337.8	174.6	217.50 (102.50–441.10)	266.5	−56.4 −126.3 ÷ −0.8	<0.001
	(%) norm *		75.2	92.57 (39.67–200.44)	118.4	51.6	68.30 (24.40–181.36)	88.7	−26.2 −47.4 ÷ −3.9	<0.001
	% E from C		46.7	50.67 (39.28–68.19)	56.7	45.5	50.46 (25.43–66.86)	55.5	−0.02 −8.2 ÷ 4.9	0.25
“Simple sugars”	(g/day)	68	54.3	81.01 (14.10–375.90)	105.7	32.2	52.05 (10.28–227.00)	65.7	−26.6 −61.9 ÷ 2.86	<0.001
	(%) norm *		65.5	105.25 (17.90–345.65)	145.2	34.7	54.77 (12.75–207.25)	97.1	−34.6 −80.2 ÷ −0.7	<0.001
	% E from SS		11.3	15.04 (4.00–32.01)	19.3	8.7	10.16 (3.38–33.28)	14.7	−2.7 −8.1 ÷ 1.9	0.0009
Dietary fibre	(g/day)	68	16.0	19.45 (8.80–36.10)	25.1	15.7	18.95 (7.90–42.70)	23.4	−0.05 −6.0 ÷ 3.9	0.52
	(%) norm *		94.5	114.47 (53.15–199.28)	134.2	82.5	102.53 (41.57–224.73)	134.2	−11.1 −31.6 ÷ 18.5	0.08

*n*—number of children; * percentage of norm realization; %E from P—percentage of energy coming from proteins; %E from F—percentage of energy coming from fat; %E from C—percentage of energy coming from carbohydrates; %E from SS—percentage of energy coming from “simple sugars”; *n*—number of children; Me—median; Q_1_, Q_3_—quartile.

**Table 6 ijerph-19-11686-t006:** The correlation between macronutrient consumption and measured somatic development parameters in children with excess body weight after one-year dietary intervention.

Measured Parameter	Waist Circumference SDS		WHtR		% of Adipose Tissue in the Body	
	*n* = 68		*n* = 68		*n* = 68	
	rs	*p*-Value	rs	*p*-Value	rs	*p*-Value
Percentage of protein consumption	0.24	0.051	0.25	0.03	0.15	0.23
Percentage of fat consumption	0.13	0.27	0.16	0.17	0.12	0.34
Percentage of saturated fat consumption	0.17	0.17	0.17	0.16	0.15	0.24
Percentage of cholesterol consumption	−0.31	0.76	−0.62	0.53	−0.40	0.68
Percentage of carbohydrate consumption	0.20	0.10	0.26	0.03	0.27	0.03
Percentage of “simple sugars” consumption	0.57	0.57	0.47	0.63	0.98	0.32
Percentage of dietary fibre consumption	−0.57	0.56	−0.23	0.82	0.60	0.54

WHtR—waist/height ratio; r_s_—correlation coefficient Spearman’s rank; SDS—standard deviation score; *n*—number of children.

**Table 7 ijerph-19-11686-t007:** The correlation between macronutrient consumption and measured parameters of lipid profile in children after dietary intervention.

Measured Parameter	Total Cholesterol		LDL-CH		HDL-CH		TG		TG/HDL Ratio	
	*n* = 67		*n* = 66		*n* = 67		*n* = 67		*n* = 67	
	r_s_	*p*-Value	r_s_	*p*-Value	r_s_	*p*-Value	r_s_	*p*-Value	r_s_	*p*-Value
Protein consumption percentage	0.16	0.2	0.28	0.02	0.12	0.3	0.03	0.8	−0.14	0.9
Fat consumption percentage	0.21	0.08	0.16	0.19	0.19	0.13	0.26	0.03	0.18	0.15
Saturated fat consumption percentage	0.08	0.5	0.15	0.22	0.21	0.08	0.10	0.4	0.06	0.63
Cholesterol consumption percentage	−0.01	0.94	0.04	0.8	0.13	0.30	0.06	0.65	0.02	0.87
Carbohydrate consumption percentage	0.06	0.63	0.02	0.85	0.03	0.81	0.12	0.33	0.10	0.41
“Simple sugars” consumption percentage	0.02	0.88	−0.06	0.65	0.05	0.68	0.14	0.24	0.09	0.44
Dietary fibre consumption percentage	0.03	0.79	0.05	0.71	0.2	0.11	0.003	0.88	−0.01	0.91

LDL-CH—LDL-cholesterol; HDL-CH—HDL-cholesterol; TG—triglycerides; r_s_—correlation coefficient Spearman’s rank; *n*—number of children.

**Table 8 ijerph-19-11686-t008:** Correlations between macronutrient consumption and measured glucose metabolism parameters in children after dietary intervention.

Measured Parameter	Fasting Glucose		Glucose at 120′ of OGTT		Fasting Insulin		Insulin at 120′ of OGTT		HOMA-IR	
	*n* = 67		*n* = 65		*n* = 66		*n* = 64		*n* = 66	
	r_s_	*p*-Value	r_s_	*p*-Value	r_s_	*p*-Value	r_s_	*p*-Value	r_s_	*p*-Value
Protein consumption percentage	0.22	0.07	0.10	0.40	0.12	0.32	0.22	0.07	0.18	0.14
Fat consumption percentage	0.14	0.24	0.16	0.20	0.16	0.21	0.23	0.06	0.18	0.16
Saturated fat consumption percentage	0.22	0.07	0.09	0.49	0.15	0.21	0.19	0.14	0.18	0.16
Cholesterol consumption percentage	0.08	0.53	0.10	0.40	0.13	0.29	0.16	0.19	0.18	0.19
Carbohydrate consumption percentage	0.29	0.01	0.24	0.054	0.17	0.16	0.35	0.04	0.22	0.07
“Simple sugars” consumption percentage	0.008	0.95	0.25	0.04	0.15	0.22	0.32	0.01	0.16	0.18
Dietary fibre consumption percentage	0.16	0.19	0.04	0.75	−0.009	0.94	0.06	0.63	0.03	0.81

OGTT—oral glucose tolerance test; HOMA-IR—insulin resistance index; r_s_—correlation coefficient Spearman’s rank; *n*—number of children.

## Data Availability

Data are available on request.

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
