# Peer review of "Impact of One-Year Dietary Education on Change in Selected Anthropometric and Biochemical Parameters in Children with Excess Body Weight"

_ijerph, 2022, doi:10.3390/ijerph191811686_

Round 1
Reviewer 1 Report
The outcomes measured as part of this study are interesting and relevant to the population of children in treatment for obesity. However I have significant concerns about the methodology and more explicit details needed to demonstrate the soundness of approaches:
e.g. what is 23h nutrition history? Is it 3*24 hour recalls or 3 day self-report food diary? At what time points were they collected? How was accuracy in food diaries accounted for? Underreporting, a significant issue, wasn't discussed in terms of interpreting food diaries.
Were food diaries used to influence dietary advice? Were any behaviour change techniques integrated into dietary counselling? Recommend using the behaviour change taxonomy to define those clearly. Insufficient explanation of how dietary intervention was planned, carried out and evaluated systematically.
The wide age range of the sample and combination of children with overweight and obesity together in analysis makes it difficult to separate out important categorical differences that we know exist.
The elimination of children who dropped out of the study from analysis is problematic and how is this accounted for in the statistical model?
Author Response
e.g. what is 23h nutrition history? Is it 3*24 hour recalls or 3 day self-report food diary? At what time points were they collected? How was accuracy in food diaries accounted for? Underreporting, a significant issue, wasn't discussed in terms of interpreting food diaries.
Detailed way of taking nutritional history was added to the methodology section:
The dietary interview was always collected in the presence of the same qualified clinical dietician in a way that ensured each surveyed person understood all the questions. The patient described 3 consecutive days of menus (including one weekend day) from before hospitalisation. In order to avoid over- or under-reporting of portions eaten by respondents, to estimate portion sizes Album of Photographs of Food Products and Dishes by National Food and Nutrition Institute was used [30]. This album is a collection of photographed products and meals in 3 different portion sizes converted into grams. The patient, after looking at the photo, indicated the closest size of the product they had used to prepare the meal or the meal itself that they had eaten. Additionally, the participants were suggested to use the website www.ilewazy.pl, “Photographic weight converter useful in cooking and dieting” [31]. This website is used as a way to recalculate traditional measurement units (such as cup, tablespoon, handful, pinch) into grams. To calculate the calorie and basic macronutrients consumption, DIETA.5 program by National Food and Nutrition Institute was used [32].
Added to discussion:
The study used a 24-hour dietary interview, which is considered one of the more accurate methods used to estimate average food intake with a questionnaire. It is characterised by a lower possibility of error concerning the omission or addition of a product or dish. This method also has the advantages of a relatively short time between food consumption and the retelling thereof, and a fairly high accuracy of the data obtained.
Were food diaries used to influence dietary advice? Were any behaviour change techniques integrated into dietary counselling? Recommend using the behaviour change taxonomy to define those clearly. Insufficient explanation of how dietary intervention was planned, carried out and evaluated systematically.
The details on the organisation of dietary education were added to the methodology section:
Educational measures were aimed at both children and their parents or carers during all visits. During the study children took part in 6 individual education meet-ings. During the first 3 month they were controlled once every three months, control 4-6 took place once every 3 months. For the 6th visit children along with their legal guardians were invited back to the Department to perform control laboratory tests, anthropometric measurements and nutrition assessment. On every visit the children and their guardians were re-educated. During the education special emphasis was placed on: reducing the intake of saturated fats in the diet, by: eliminating high-calorie snacks such as crisps, nuts, salty sticks and fast-foods; limiting fried meals; increasing the consumption of fish, lean meat and dairy products; increasing the amount of fruit and vegetables consumed; limiting the consumption of sugar, sweets, sweetened dairy products; eliminating sweetened drinks and fruit juices; introducing whole grain cereal products; drinking water daily; eating an adequate amount of food regularly; making conscious decisions to choose healthy foods - being able to analyse product labels; en-couraging daily physical activity. Children and parents/carers were encouraged to eat together at the table and avoid eating meals and snacks in front of the TV, computer or smartphone. The current nutrition diary was analysed, all the errors were corrected. Easy, cheap and healthy ways of reducing the caloric value of consumed meals were proposed. Children and their parents were motivated to together change the incorrect nutrition habits. During the follow-up meetings, special attention was paid to the op-portunity to ask questions and ensuring that sufficient time was given for each visit (40-60 minutes on average). The study presented the children's parents with an op-portunity for one-on-one meetings with an experienced dietitian, giving greater op-portunity to focus on the child's personal problems, preferences and dietary predispo-sitions. Patients also had the opportunity to send questions to the educator via e-mail and telephone text messages. Most importantly, respondents also had the opportunity to adjust their individual appointment from among the 3 proposed dates, thus avoid-ing missed appointments.
The wide age range of the sample and combination of children with overweight and obesity together in analysis makes it difficult to separate out important categorical differences that we know exist.
Of course combining children with overweight and obesity in one group could be considered a certain limitation, which in our study is also caused by the sample size of the intervention group.
An additional sentence was added to methodology section: Due to social reasons, participation in the project was proposed to all children with obesity and overweight diagnosed in the Clinic.
The elimination of children who dropped out of the study from analysis is problematic and how is this accounted for in the statistical model?
In terms of anthropometric parameters, children who completed the dietary in-tervention did not differ from children who dropped out of the intervention. The data is now presented in Table 1.
Reviewer 2 Report
This is an interesting manuscript that shows the effect of a dietary intervention on weight and metabolic parameters. However, in order to be accepted, it must address the following suggestions:
1. Did the Authors verify dietary intake for plausible reporters? Under- or over-reporting in children and adolescents is common, specially among those with overweight/obesity. Results must be adjusted for this variable as a confounder at baseline.
2. Lines 132-136: What cut-off points were considered as abnormal blood pressure?
3. I would like to know the compliance of dietary intervention from the sample. Did all participants attend to the scheduled appointments for dietary intervention?
4. Please verify decimals in methods and results sections(sometimes are marked as commas), change accordingly to the journal's guidelines.
5. Change p-values written as <0.00001 to "p<0.001"
6. Improve Tables´ presentation, some lines are marked as bold. Verify decimal markers.
7. In table 1, as mentioned before, change the comma for decimals and provide the p-value for Adipose tissue.
8. Table 2. Improve presentation and provide p-values where "NS" is stated.
9. Table 3. Must be improved, lines are missing and again, commas are used for decimals. Homa IR difference value is wrong.
10. Table 4. Did the Authors verify the distribution of dietary intake variables? Energy intake and other nutrients often do not have a normal distribution. Therefore, to show Mean and SD for those variables is not correct. Means and interquartile ranges must be estimated for skewed variables.
11. Table 5. Add p-value for "NS" variables. Change commas in decimals.
12. „Simple sugars” has a mistype error along the manuscript, please correct.
13. Table 6. Improve presentation and use decimal markers according to journal´s guidelines. Although some correlations are significant, the correlation coefficient is low. This should be stated in discussion section.
14. There is a mistype error in line 352 "correaltion", please correct.
15. Improve design and presentation for Table 7, some numbers are in red. Provide p-value for "NS"
16. Did sex or age had any interaction with results? All tables are provided for full sample, but, for example children at the age of 3 have a totally different energy and nutrient intake than adolescents. Thus, I consider these variables should be treated with caution and present results by age- or sex- stratification.
17. Also, correlations must be adjusted for sex, age and tanner stages for adolescents.
Author Response
1.Did the Authors verify dietary intake for plausible reporters? Under- or over-reporting in children and adolescents is common, specially among those with overweight/obesity. Results must be adjusted for this variable as a confounder at baseline.
A detailed explanation was added to the methodology section.
2.Lines 132-136: What cut-off points were considered as abnormal blood pressure?
Thank you for this important remark. The norms and cut-off points description was added to the methodology section:
To assess the frequency of abnormal arterial blood pressure values in the study group, values <90th percentile for sex, age and height percentile were considered normal. Values between the 90th and 95th percentile were defined as high normal, and a values ≥95th percentile for sex, age and height percentile were considered as elevated blood pressure.
3.I would like to know the compliance of dietary intervention from the sample. Did all participants attend to the scheduled appointments for dietary intervention?
Wider explanation of compliance was added to the methodology section:
The study presented the children's parents with an opportunity for one-on-one meet-ings with an experienced dietitian, giving greater opportunity to focus on the child's personal problems, preferences and dietary predispositions. Patients also had the op-portunity to send questions to the educator via e-mail and telephone text messages. Most importantly, respondents also had the opportunity to adjust their individual appointment from among the 3 proposed dates, thus avoiding missed appointments.
4.Please verify decimals in methods and results sections(sometimes are marked as commas), change accordingly to the journal's guidelines.
All the decimals were changed according to the journal’s guidelines.
5.Change p-values written as <0.00001 to "p<0.001"
p-values notation was changed
6.Improve Tables´ presentation, some lines are marked as bold. Verify decimal markers.
Tables presentation was unified, decimals were changed according to the journal’s guidelines.
7.In table 1, as mentioned before, change the comma for decimals and provide the p-value for Adipose tissue.
Decimals were changed according to the journal’s guidelines. P-value for adipose tissue was added.
- Table 2. Improve presentation and provide p-values where "NS" is stated.
Table’s presentation was improved, p-values were provided for non-significant p values
- Table 3. Must be improved, lines are missing and again, commas are used for decimals.
Homa IR difference value is wrong.
Table 3 was graphically corrected, decimals were changed. HOMA IR difference value was corrected.
- Table 4. Did the Authors verify the distribution of dietary intake variables? Energy intake and other nutrients often do not have a normal distribution. Therefore, to show Mean and SD for those variables is not correct. Means and interquartile ranges must be estimated for skewed variables.
The distribution was verified by our statistician. Interquartile ranges were added to the table.
- Table 5. Add p-value for "NS" variables. Change commas in decimals.
P-values were added for all variables. Decimals were corrected according to the journal’s guidelines.
- „Simple sugars” has a mistype error along the manuscript, please correct.
Mistypes were corrected.
- Table 6. Improve presentation and use decimal markers according to journal´s guidelines. Although some correlations are significant, the correlation coefficient is low. This should be stated in discussion section.
The presentation and decimals were corrected according to the journal’s guidelines. The weakness of the correlations was specified were necessary in the discussion.
14.There is a mistype error in line 352 "correaltion", please correct.
Mistype error corrected.
15.Improve design and presentation for Table 7, some numbers are in red. Provide p-value for "NS"
The table was graphically corrected, with red color being eliminated. P-values were provided for all variables.
16.Did sex or age had any interaction with results? All tables are provided for full sample, but, for example children at the age of 3 have a totally different energy and nutrient intake than adolescents. Thus, I consider these variables should be treated with caution and present results by age- or sex- stratification.
To eliminate the effect of sex and age for a small and non-homogenous group for most anthropometric parameters standard deviation score (SDS) values were calculated, using referential values for the population of Polish children.
Due to wide age range within the group, caloric demand was individually calculated for every child using the norms by Institute of Food and Nutrition from 2012. Later, for every child the percentage of their norm realisation was calculated for energy, protein, total fat, saturated fats, cholesterol, carbohydrates, “simple sugars” and dietary fibre.
17.Also, correlations must be adjusted for sex, age and tanner stages for adolescents.
Unfortunately, the small sample size does not allow for additional analyses of achieved results with inclusion of age and Tanner stage. Most of the parameters associated with the calculation of caloric and macronutrient demand were calculated individually for every child using the Institute of Food and Nutrition 2012 norms.
Reviewer 3 Report
The authors assessed the effectiveness of a one year dietary intervention (nutrition pyramid) in children and adolescents with excess body weight. The manuscript is well-written with appropriate methodology. The results were clearly described and discussed. This article will be of interest to readers.
Author Response
Thank you kindly for the very positive review. We are very happy to have caught your interest.
Reviewer 4 Report
In this study, the authors sought to evaluate the effect of a one-year dietary education intervention on anthropometric parameters – body weight, waist-to-hip ratio, adipose and lean tissue – and biochemical parameters - lipid profile, fatty liver, and blood pressure – in overweight and obese children. They reported a decrease in BMI, W-t-H ratio, and adipose tissue. They also reported had a more balanced diet, in term of their macronutrient intake, after the intervention.
The authors did a great job describing the objectives of the study; however, I think that the anthropometric data should be reported in absolute values not in standard deviation scores. Reporting absolute values may introduce the need to divide the subjects by age, but it will allow the reader to better gauge the impact of the intervention.
It is also important to include a short description of the educational level of the parents of the children that participated in the study and their involvement in this process. As written, it makes the impression that the subjects were the ones being educated and making choices for themselves when in fact the parents and/or caregivers were the ones making choices for the most part.
Author Response
The authors did a great job describing the objectives of the study; however, I think that the anthropometric data should be reported in absolute values not in standard deviation scores. Reporting absolute values may introduce the need to divide the subjects by age, but it will allow the reader to better gauge the impact of the intervention.
Due to the wide age range of children in the study group, SDS was used in place of absolute values. The authors believe it is more appropriate in the context of significantly different ages of children within the study group.
It is also important to include a short description of the educational level of the parents of the children that participated in the study and their involvement in this process. As written, it makes the impression that the subjects were the ones being educated and making choices for themselves when in fact the parents and/or caregivers were the ones making choices for the most part.
A description of the parents’ education level was added to the result section:
The education level of the studied children's mothers varied. Almost half (47.8%) of them had secondary education, more than a third (35.8%) had higher education, and primary or vocational education respectively 8.9% and 7.5%. The vast majority of mothers (83.6%) were professionally active. In the study group, 2/3 (64.2%) of the mothers lived in urban areas and 1/3 (35.8%) in rural areas. Nearly 3/4 (68.7%) of the mothers were also characterised by excessive body weight. It was observed that those who dropped out of the study were characterised by a "lower degree" of motivation to change their children's eating habits and reduce their weight, compared to those who remained in the study until its completion. This may suggest that the patients who remained in the study were characterised by a higher-than-average degree of motiva-tion to lose weight, which translated into their weight reduction outcomes. The need for the children and their parents to dedicate a huge amount of time during frequent follow-ups over the 12-month dietary intervention period was also an obstacle.
A clarification concerning educational intervention was added to the methodology section:
Educational measures were aimed at both children and their parents or carers during all visits.
Round 2
Reviewer 2 Report
The authors made a substantial revision of the manuscript. However, in order to be accepted, minor changes must be addressed:
1) Tables must show either mean (SD) or median (Interquartile range) for variables (according to normality test). This recommendation includes biochemical parameters.
2) Table 6 (correlation) must show p-values for all measurements.
3) If adjustment for potential confounders could not be made due to limited sample size, please add to limitations in the discussion section.
Author Response
- Tables must show either mean (SD) or median (Interquartile range) for variables (according to normality test). This recommendation includes biochemical parameters.
In table 4 mean values and SD were removed, leaving medians, quartiles, minimal and maximal values, to keep one way of presenting results.
- Table 6 (correlation) must show p-values for all measurements.
In table 6 p-values were presented for all the parameters described.
- If adjustment for potential confounders could not be made due to limited sample size, please add to limitations in the discussion section.
A comment on limitations caused by sample size was added in discussion.